# Divergent molecular pathways govern temperature-dependent wheat stem rust resistance genes

Tim C. Hewitt [1,8,10], Keshav Sharma[2,10], Jianping Zhang [3,4,10], Chunhong Chen[1], Prabin Bajgain[5], Dhara Bhatt[1], Smriti Singh[3], Pablo D. Olivera Firpo[2,6], Jun Yang[1], Qiaoli Wang[4], Narayana Upadhyaya [1], Curtis Pozniak [7], Robert McIntosh[3] ✉, Evans Lagudah [1,3] ✉, Peng Zhang [3] ✉ & Matthew N. Rouse [2,9] ✉

The wheat stem rust pathogen *Puccinia graminis* f. sp. *tritici* (*Pgt*) causes severe crop losses worldwide. Several stem rust resistance (*Sr*) genes exhibit temperature-dependent immune responses. *Sr6*-mediated resistance is enhanced at lower temperatures, whereas *Sr13* and *Sr21* resistances are enhanced at higher temperatures. Here, we clone *Sr6* using mutagenesis and resistance gene enrichment and sequencing (MutRenSeq), identifying it to encode a nucleotide-binding leucine-rich repeat (NLR) protein with an integrated BED domain. *Sr6* temperature sensitivity is also transferred to wheat plants transformed with the *Sr6* gene. Differential gene expression analysis of near-isogenic lines inoculated with *Pgt* at varying temperatures reveals that genes upregulated in the low-temperature-effective *Sr6* response differ from those upregulated in the high-temperature-effective responses associated with *Sr13* and *Sr21*. These findings highlight divergent molecular pathways involved in temperature-sensitive immunity and inform future strategies for deployment and engineering of genetic resistance in response to a changing climate.

Wheat stem rust, caused by the fungal pathogen *Puccinia graminis* f. sp. *tritici* (*Pgt*), poses a significant threat to global wheat production. Virulent strains have overcome widely deployed disease resistance genes, including *Sr6*[1]. This gene, once widely effective, is now widespread and found in ~13% of global spring wheat varieties[2]. Stem rust epidemics have historically caused severe damage during warm weather, affecting continental-scale wheat production[3,4]. In the United States, stem rust outbreaks from 1919 to 1954 led to substantial statewide losses[5], often causing up to 50% yield loss[3].

The release of the *Sr6*-bearing cultivar Selkirk was particularly successful in combating stem rust epidemics that swept across North America from the 1920s to 1960s. The first widely grown Australian stem rust-resistant cultivar was Eureka, carrying *Sr6*, and released in 1938. However, as early as 1942, virulent *Pgt* races were identified, and

[1]CSIRO Agriculture & Food, Canberra, ACT, Australia. [2]USDA-ARS, Cereal Disease Laboratory, St. Paul, MN, USA. [3]University of Sydney, Plant Breeding Institute, School of Life and Environmental Sciences, Cobbitty, NSW, Australia. [4]Centre for Crop Genome Engineering and College of Agronomy, Longzi Lake Campus, Henan Agricultural University, Zhengzhou, China. [5]Department of Agronomy and Plant Genetics, University of Minnesota, St. Paul, MN, USA. [6]Department of Plant Pathology, University of Minnesota, St. Paul, MN, USA. [7]Crop Development Centre and Department of Plant Sciences, University of Saskatchewan, Saskatoon, SK, Canada. [8]Present address: Immunology & Infectious Diseases, John Curtin School of Medical Research, Australian National University, Canberra, ACT, Australia. [9]Present address: USDA-ARS, Sugarcane Field Station, Canal Point, FL, USA. [10]These authors contributed equally: Tim C. Hewitt, Keshav Sharma, Jianping Zhang. ✉e-mail: robert.mcintosh@sydney.edu.au; evans.lagudah@csiro.au; peng.zhang@sydney.edu.au; matthew.rouse@usda.gov

the cultivar went through a series of recurrent epidemics before it was removed from cultivation after the mid-1960s[6]. Nevertheless, *Sr6* still provides resistance in some parts of North America[7,8] and possibly India[9], particularly when combined with other genes such as *Sr57* (*Lr34*)[10] or *Sr2* (as was the case with Selkirk).

The emergence of *Pgt* races TTKSK (commonly referred to as Ug99) and TKTTF ('Digalu' race) in East Africa was particularly alarming. These races were virulent to widely deployed stem rust resistance genes, rendering 80–90% of global wheat varieties susceptible[11–15]. Spread across multiple regions, the Ug99 and Digalu races and variants overcame resistance genes such as *Sr24*, *Sr36*, and *SrTmp*[11–13,16].

Plants employ basal and resistance (R)-mediated defense responses to infection by microbial pathogens[17]. Both responses are influenced by heat stress[18], highlighting the adaptability of the plant defense system to varying temperatures. Pathogen-associated molecular patterns (PAMPs) activate PAMP-triggered immunity (PTI) upon detection. Host-adapted pathogens manipulate PTI using effector proteins, which can be recognized by plant intracellular receptors, leading to Effector-Triggered Immunity (ETI). ETI, mediated by host *R* genes, results in hypersensitive responses or programmed cell death[18].

Durable disease resistance is essential to safeguard wheat from ever-changing rust pathogens[19]. Stem rust resistance is often placed in two categories: Adult plant resistance (APR), conferred by multi-pathogen resistance genes such as *Sr2*, *Sr57/Lr34/Yr18*, *Sr58/L46/Yr29*, and *Sr55/Lr67/Yr46* (and other designations) that show effectiveness at later growth stages, and all-stage resistance (ASR), effective from the seedling stage. Cloned ASR genes for stem rust include *Sr22, Sr26, Sr27, Sr33, Sr35, Sr43, Sr45, Sr46, Sr50, Sr60, Sr61, Sr62* and *Sr66* as well as the temperature-sensitive *Sr13* and *Sr21*[20–27].

These *R* genes typically encode coiled coil-nucleotide binding leucine-rich repeat (CC-NBS-LRR or NLR) proteins or protein kinases, such as *Sr60*[27]. Notably, *Sr13* and *Sr21* are more effective at higher temperatures[1,11,21,26,28], in contrast to *Sr6,* which is more effective below 20 °C and ineffective above 24–27 °C[29,30]. Similarly, stem rust resistance genes *Sr10, Sr15*, and *Sr17* are less effective at higher temperatures[21]. Plants susceptible at a non-permissive temperature regained resistance once the temperature returned to a permissive level[31]. Light conditions also impact the resistance response[32].

Whereas *Sr13* and *Sr21* are previously cloned[21,26], here we report the cloning of *Sr6* using mutagenesis and *R* gene enrichment and sequencing (MutRenSeq) and show that it is an NLR integrated with a non-canonical zinc finger BED domain. Stable transformation confirms *Sr6* identity, which induces a characteristic temperature-sensitive resistance phenotype in transgenic plants challenged with an *Sr6*-avirulent *Pgt* race. RNA sequence analysis shows that gene expression is affected by temperature. Differential gene expression analysis on near-isogenic wheat lines carrying either *Sr6, Sr13,* or *Sr21* elucidates varied defence pathways in response to different temperatures.

## Results

### *Sr6* candidate isolated by MutRenSeq and verified in recombinant inbred lines

To clone *Sr6*, we identified susceptible ethyl methanesulfonate (EMS)-generated mutants from the substitution line Chinese Spring*5/Red Egyptian 2D (CS/RE 2D) produced by Sears et al.[33]. Seven independent mutants, together with wild-type CS/RE 2D, were processed using the MutRenSeq pipeline. Captured sequencing reads from these seven mutants and the wild type (WT) were aligned to a de novo reference assembly of the WT reads. One contig (#733836) of 2842 bp containing a SNP was identified in three of the seven mutants and possessed NB-ARC and LRR motifs but no CC motif. A second contig (#737511) of 2218 bp containing a SNP in a different three mutants was identified and possessed only LRR motifs. The last ~600 bp of contig #733836 had homology to the first ~600 bp of contig #737511, suggesting they overlapped but were not joined during assembly, possibly due to low

coverage and/or ambiguity of reads covering the bridging region. To confirm that these sequences were physically joined, a second assembly was generated based on a combined pool of reads from all the mutants and WT. The second assembly produced a scaffold (#2265) that represented a joining of the two contigs in six of the seven mutants having SNPs (Supplementary Fig. 1A).

No SNP was detected in mutant 3981-4, but since scaffold #2265 contained no CC domain-encoding motifs, mutant 3981-4 was predicted to harbor a mutation in a potentially missing upstream sequence. This prediction was later confirmed when Sanger sequencing identified a SNP within the 5′UTR of 3981-4 (Fig. 1A). Scaffold #2265 was aligned to the Chinese Spring RefSeq v1.0 (CSv1) reference assembly[34] using BLAST, with the top hit (~91%) to chromosome 2D approximately 5.1 Mbp from the *Sr6*-proximal marker *Xwmc453* identified by Tsilo et al.[32]. The matching sequence overlapped with high-confidence annotated gene *TraesCS2D02G111500*, encoding a disease resistance protein. A dominant PCR marker *Sr6STS1* designed based on the sequence of scaffold #2265 showed specificity to *Sr6*-carrying lines, CS/RE 2D, and Manitou (Supplementary Fig. 2A).

Additionally, a PCR product spanning almost the full length of scaffold #2265 confirmed linkage of the two initially discovered contigs (Supplementary Fig. 2B) and was amplified from the six mutants with SNPs. Sanger sequencing confirmed the presence of the SNP mutations. Each of the six mutants contained at least one nonsynonymous SNP, with five resulting in missense mutations (mutants 3704-6, 3706-7, 4002-6, 4190-6, and 5047-4) and one resulting in a nonsense mutation (mutant 5140-5) (Fig. 1A). 5′RACE (rapid amplification of cDNA ends) from WT RNA revealed the presence of an additional 5′ sequence that encoded both CC and zinc finger BED domains. Primers were designed to amplify this additional 5′ region from the DNA of mutant 3981-4 (Supplementary Fig. 2C), from which a mutation had yet to be identified. Sanger sequencing revealed the presence of a G-to-A SNP (Supplementary Fig. 1B) in the 5′UTR, 20 bp upstream of the first start codon. Although this mutation does not alter the coding sequence, mutations in the 5′UTR are known to impact translation[35,36]. Overall, point mutations were confirmed in the candidate sequence for all seven mutants (Supplementary Table 1).

To further verify the candidate as *Sr6*, a population of 197 $F_3$ lines from a reciprocal cross of CS and CS/RE 2D segregating for *Sr6* was screened for response to *Pgt* race 21-2,3,7 (University of Sydney Culture no. 216). The dominant STS marker *Sr6STS1*, based on the candidate sequence, was present in all homozygous resistant and segregating lines, and absent in homozygous susceptible lines.

### Full gene structure of *Sr6* obtained from whole-genome sequenced wheat accessions

The *Sr6* candidate sequence compared against published wheat genome assemblies[37] had a 100% match on chromosome 2D of the genome assembly of cv. Landmark (Supplementary Fig. 3). The full upstream and downstream sequences from Landmark presented an opportunity to use the native regulatory elements for transgenic experiments with *Sr6*. However, the probable promoter region, less than 2 kb upstream of the putative start of codon, contained a large gap in sequence denoted by Ns (unknown bases). Fortunately, the *Sr6* sequence was also found in an assembled scaffold of cv. Claire that harbors a gapless upstream sequence. The sequence from Claire was used to close the gap present in the Landmark sequence (Supplementary Fig. 4A). Additionally, a scaffold from cv. Robigus, which contained a non-identical coding sequence, did have an identical upstream region to Claire, indicating that the sequence overlapping the gap was conserved.

A transcript was identified from 5′ and 3′ RACE amplification, having an intron/exon structure showing conservation with the CSv1 homolog *TraesCS2D02G111500*. Additionally, a complete transcript retrieved from the de novo transcriptome assembly of Avocet R

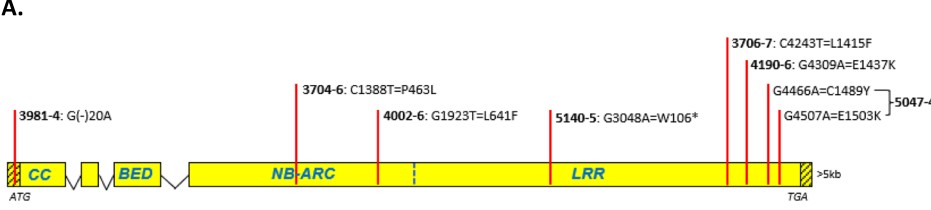

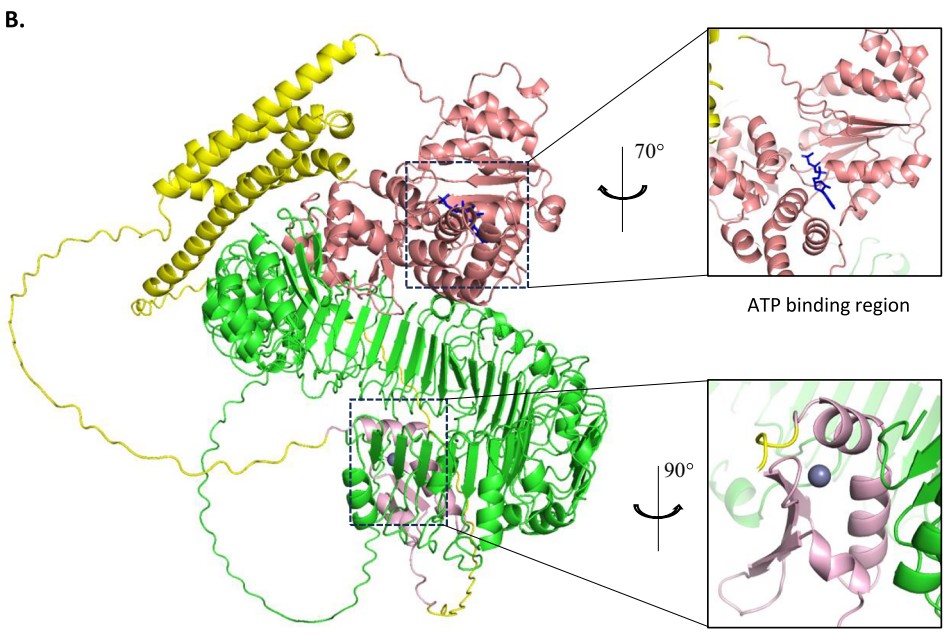

ATP binding region

Zinc ion binding region

**Fig. 1 | *Sr6* gene and SR6 protein structures. A** *Sr6* gene structure showing positions of EMS-induced SNPs identified in mutants. SNP positions are shown by red bars labelled with mutant number, and altered amino-acid positions are given. Shaded areas are UTRs. Domain encoding regions are indicated: CC coiled-coil, NB-ARC nucleotide binding (APAF-1, R proteins and CED-4), LRR leucine-rich repeat. **B** Full-length structure of SR6 protein, its zinc ion binding region, and ATP binding region are predicted by AlphaFold3. The full-length structure comprises three subdomains: the BED motif (light pink shaded) containing the N-terminal subdomain (yellow shaded), the NB-ARC domain (salmon shaded), and the LRR domain (green shaded).

supported conservation of the intron/exon structure at the 5' end (Supplementary Fig. 3).

### *Sr6* is diverged from known wheat NLRs but contains a conserved BED domain

The *Sr6* candidate was aligned to 16 reference genomes, of which only Landmark and Claire contained a matching sequence. However, the ~170 bp BED domain encoding region (exon 2) exhibited higher conservation, as seen in the alignment with the Avocet R ortholog (Supplementary Fig. 3). The BED domain sequence was also identical in chromosome 2D homologs in ArinaLrFor, Jagger, Julius, Lancer, Norin 61, and Robigus. The BED domains of the 2D homologs in CSv1 and Weebill 1, as well as the chromosome 2B homoeolog in Zavitan (tetraploid) had eight SNPs with *Sr6*. The 2D homologs in Mace, SY Mattis, and spelt accession PI190962 had 16 SNPs.

The full-length SR6 protein, its zinc ion binding region and ATP binding region were predicted by AlpahFold3[38] (Fig. 1B). To date, six BED-NLRs associated with pathogen resistance have been isolated, including *Yr5/YrSP*, *Yr7* (resistance to wheat stripe rust)[39], *Xa1* and *Xo1* (resistance to rice blight and leaf streak)[40,41], *Rph15* (resistance to barley leaf rust)[42], and *Pm6Sl* (resistance to powdery mildew)[43]. The BED domain of *Sr6* was compared with these six genes, revealing all seven to have the same CC-BED module at their N-terminus, suggesting a conserved structure configuration in BED-NLRs. Amino acid sequence

alignment showed that they share four BED motifs considered to be conserved in land plants[44], along with an additional four residues common across all seven sequences (Supplementary Fig. 4B), potentially indicating functional conservation among BED-NLRs associated with resistance. While the full SR6 protein is highly divergent from the other six BED-NLRs, it appears most similar to RPH15 from barley (Supplementary Fig. 4C).

The translated sequence of the *Sr6* candidate was compared against the NCBI protein database but produced no significant hits (<80% sequence identity). Comparison of the protein sequence of the candidate against a panel of known CNL class NLRs did not indicate any close relationship although it did cluster with other wheat *R* genes including *Sr21*, *Yr5*, and *Yr7* (Supplementary Fig. 5). A commonality between these three genes and *Sr6* is that they reside on group 2 chromosomes, but they are not located at orthologous positions as *Sr6* is in the 2DS arm whereas *Sr21* and *Yr5/Yr7* are in the 2AL and 2BL arms, respectively[21,39].

### Transgenic complementation with the *Sr6* candidate confers temperature-sensitive resistance in T$_1$ and T$_2$ generations

In addition to CS/RE 2D and Manitou, which are known to have *Sr6*[1], several transformable varieties were tested with marker *Sr6STS1* to screen for an appropriate transgene host. Wheat varieties Fielder and

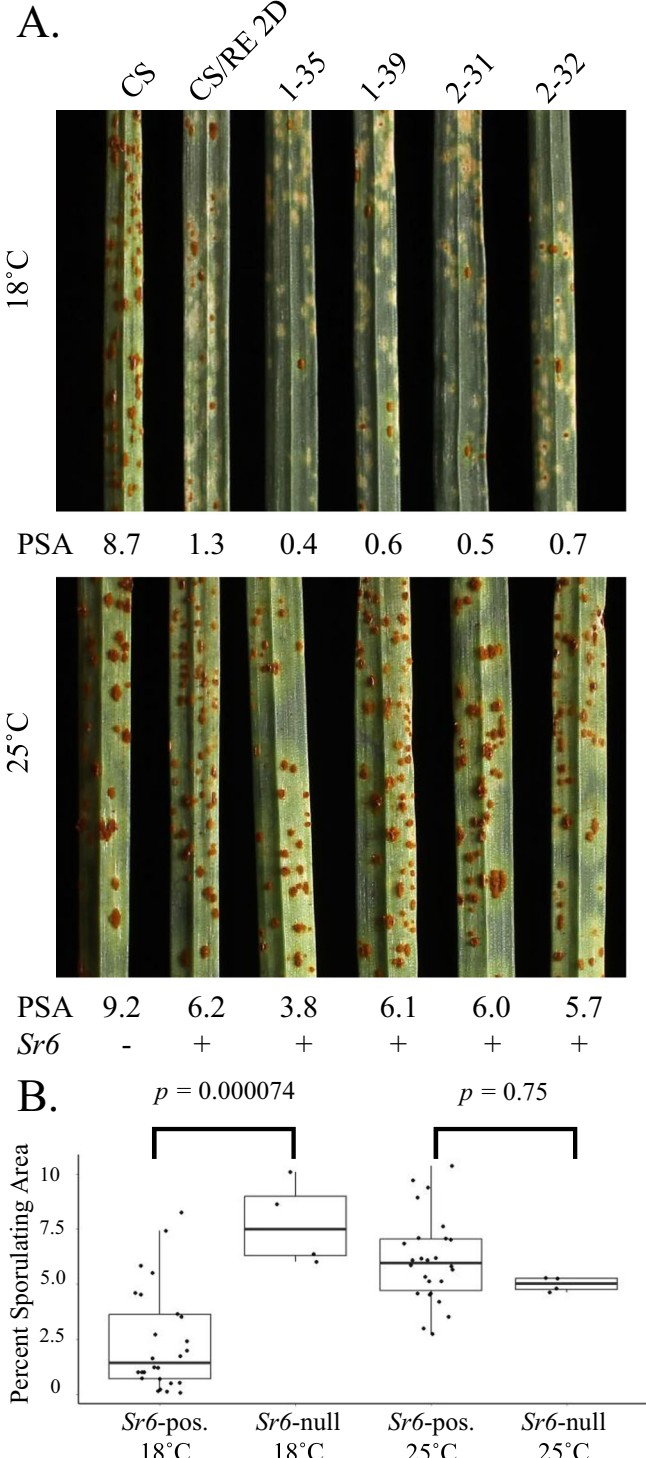

A.

PSA  8.7  1.3  0.4  0.6  0.5  0.7

PSA  9.2  6.2  3.8  6.1  6.0  5.7
*Sr6*  -    +    +    +    +    +

B.

**Fig. 2 | Westonia+*Sr6* transgenic T₂ seedling leaves at 14 days post inoculation with *Pgt* race 21-0. A** Numbered individuals were derived from either T₀ PC326-1 or PC326-2 and grown at 18 °C (A, upper panel) or 25 °C (A, lower panel). Only some representative leaves are shown. *Sr6*+ control: CS/RE 2D; *Sr6*− control: CS. Numbers at the bottom are percent sporulating areas (PSA) after images were processed with ASSESS. + or − indicate presence (+, $n = 26$) or absence (−, $n = 4$) of *Sr6* as determined by the genotypic (*Bar* gene) and phenotypic data. **B** Mean PSA was compared between all transgenic progeny (biological replicates each represented by single inoculated first leaves from different families) with or without *Sr6* at both 18 °C and 25 °C using Tukey's HSD (two-sided) (18 °C effect 5.38, 95% confidence interval 2.41 to 8.35; 25 °C effect −1.11, 95% confidence interval −4.09 to 1.86). Boxplots denote median values with second and third quartiles and with whiskers extending to the most extreme values, but no further than 1.5 times the interquartile range. Source data are provided as a Source Data file.

grown at either low (18°C) or high (25°C) temperatures and were infected with the *Pgt* race 21-0 isolate at two weeks old. The plants were scored for their resistance responses at 14 days post inoculation (dpi). At low temperature, resistance was observed as small necrotic lesions or small uredinia surrounded by necrosis. There was a clear distinction between resistant ($n = 30$) and susceptible ($n = 8$) individuals derived from T₀ PC326-1 (~3:1 segregation, $\chi^2 = 0.316$, $p = 0.574$) and a segregation of 19 resistant: 11 susceptible derived from T₀ PC326-2 ($\chi^2_{3:1} = 2.178$, $p = 0.140$), both fitting single gene segregation ratios. The presence of resistant T₁ plants confirmed resistance activity of the transgene and its identity as *Sr6*. At high temperature, all T₁ plants were susceptible, which comports with an expected loss of resistance seen with endogenous *Sr6* at high temperature and suggests temperature sensitivity was mediated by the *Sr6* gene per se.

All T₁ individuals were assayed with the *Bar* gene, a selectable marker in the *Sr6* transgene construct to check for co-segregation with resistance. Without exception, the presence of *Bar* was associated with the *Sr6* resistance phenotype, i.e., resistant plants showed *Bar* gene amplification, whereas susceptible plants did not.

T₂ populations derived from T₁ PC326-1 tested at low temperature segregated homozygous 13 resistant: 15 segregating: 8 homozygous susceptible (two resistant T₁ plants did not survive), fitting a single gene segregation ratio ($\chi^2_{1:2:1} = 2.389$, $p = 0.303$). Similarly, the T₂ population from T₁ PC326-2 segregated for 5 homozygous resistant: 13 segregating: 11 homozygous susceptible (one resistant T₁ plant did not survive), also fitting a single gene segregation ratio ($\chi^2_{1:2:1} = 2.793$, $p = 0.247$). At high temperature, all T₂ plants were susceptible (Fig. 2A, Supplementary Fig. 6). Percent sporulating areas (PSA) were compared between all transgenic progeny with versus without *Sr6* at both 18 °C and 25 °C using Tukey's HSD at $p < 0.05$. The pooled difference between *Sr6* positive and null plants at 18 °C was highly significant ($p = 0.000074$), whereas no significant difference was observed at 25 °C ($p = 0.75$) (Fig. 2B).

## *Sr* gene expression tracks with resistance response at different temperatures

LMPG and NILs, LMPG-*Sr6*, LMPG-*Sr13*, and LMPG-*Sr21*, were inoculated with *Pgt* race MCCFC (isolate no 59KS19) and sampled at different time points. LMPG was susceptible at both temperatures (Fig. 3A, B). The *Sr6* NIL was highly resistant (IT '0') at low temperature (18 °C) but susceptible (IT '3+C') at high temperature (25 °C) (Fig. 3G, H). The *Sr13* and *Sr21* NILs showed the opposite pattern and were more resistant at the higher temperature (IT '2' and '2-', respectively) compared to the low temperature (IT '2+3' and '2+', respectively) (Fig. 3C–F).

*Sr6* was upregulated at low temperature (Supplementary Fig. 7). Specifically, the level of *Sr6* expression at day 3 was higher at low temperature and lower at high temperature. At low temperature, *Sr13*

Yitpi were positive, whereas Westonia and durum cultivar Stewart were negative (Supplementary Fig. 2D, E). Furthermore, phenotyping the seed stocks of Fielder maintained at CSIRO Canberra identified low infection types (ITs) with *Sr6*-avirulent *Pgt* races 21-0 (University of Sydney Culture no. 330) and 21-2,3,7, and high IT with *Sr6*-virulent race 21-1,2,3,5,6,(7) (University of Sydney Culture no. 50); Westonia had high ITs with all three races even though it was previously reported to carry *Sr6*[45]. Consequently, the CSIRO stock of Westonia was chosen as the susceptible host for transgenic complementation.

Agrobacterium transformation of embryos of Westonia with the *Sr6* candidate gene, including native regulatory elements, yielded two independent T₀ transformants. The T₁ progeny were

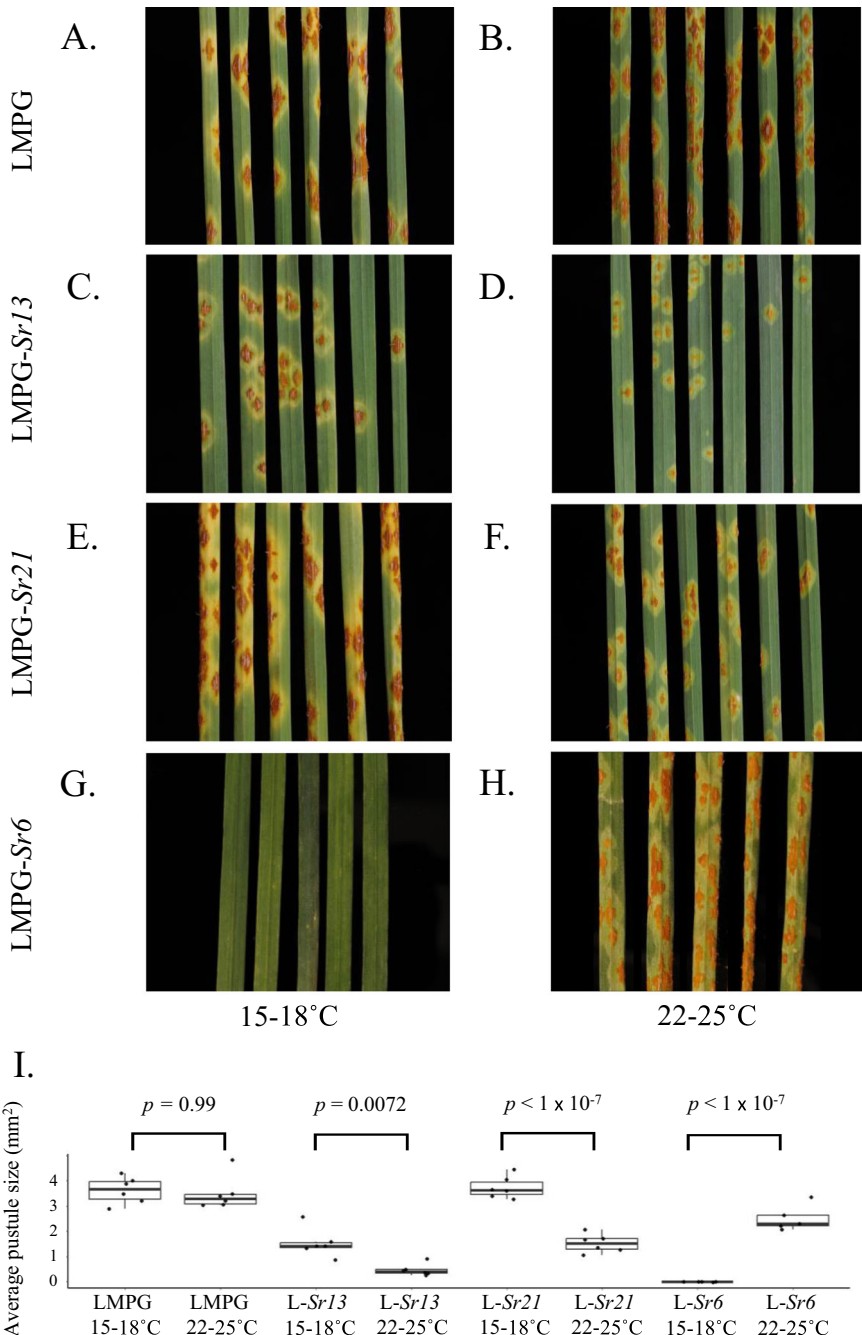

**Fig. 3 | Seedling responses of wheat lines with different host resistance genes to** *Puccinia graminis* **f. sp.** *tritici* **under high and low temperature regimes.** Biological replicates of **A** LMPG at 15–18 °C (*n* = 6), **B** LMPG at 22–25 °C (*n* = 6), **C** LMPG-*Sr13* at 15–18 °C (*n* = 6), **D** LMPG-*Sr13* at 22–25° (*n* = 6), **E** LMPG-*Sr21* at 15–18 °C (*n* = 6), **F** LMPG-*Sr21* at 22–25 °C (*n* = 6), **G** LMPG-*Sr6* at 15–18 °C (*n* = 5), **H** LMPG-*Sr6* at 22–25 °C (*n* = 5), and **I** differences of means of leaf average pustule sizes of wheat lines (**A**–**H**) with comparisons of the same line across the two temperatures using Tukey's HSD (two-sided) (LMPG effect −0.13, 95% confidence interval −0.98 to 0.73; *Sr13* effect −1.06, 95% confidence interval −1.92 to −0.20; *Sr21* effect −2.22, 95% confidence interval −3.08 to −1.36; and *Sr6* effect 2.52, 95% confidence interval 1.58–3.46). Boxplots denote median values with second and third quartiles and with whiskers extending to the most extreme values, but no further than 1.5 times the inter-quartile range. Source data are provided as a Source Data file.

was upregulated at day 1 but downregulated at day 3, whereas it was downregulated at day 1 and upregulated at day 3 at high temperature. *Sr21* was upregulated on both day 1 and day 3 under both low and high temperature conditions.

### Biotic stress-related genes show differential expression between *Sr* lines at high and low temperatures

Previous studies demonstrated involvement of *pathogenesis related* (*PR*) genes with the function of *Sr13* and *Sr21* where *PR* genes were upregulated in lines with the respective *Sr* genes and in the presence of the pathogen, but not in mutant lines without the functional *Sr* gene or in the absence of the pathogen[21,26]. Under conditions leading to more resistant responses (high temperature for *Sr13* and *Sr21*; low temperature for *Sr6*), six *PR* genes and four additional jasmonic acid (JA) pathway genes were tested by qPCR for gene expression (Supplementary Fig. 8–10, Supplementary Data 1). The *PR* genes included salicylic acid (SA) responsive genes *PR1*, *PR2*, and *PR5* in addition to JA-responsive genes *PR3* and *PR4*[26,46]. All were upregulated in *Sr6* lines at

low temperature, except *PR4,* which was downregulated at day 3. *PR1* and *PR9* were highly expressed at day 3 in *Sr6* lines. Most of the *PR* genes were upregulated in *Sr13* lines under high temperature except *PR3, PR4* and *PR5* which were downregulated at day 3 (Supplementary Fig. 9A). In *Sr21* lines at high temperature, *PR3* and *PR4* were downregulated at day 1 whereas *PR1, PR2, PR5* and *PR9* were upregulated (Supplementary Fig. 10A). All *PR* genes were downregulated in *Sr21* lines at day 3. The four additional genes associated with the JA pathway based on gene ontology (GO) were TraesCS7B02G107700-JA1, TraesCS2D02G378400-JA3, TraesCS2D02G198200-JA4, and TraesCS7A02G508800-JA6, and are described in Supplementary Data 2. As with the *PR* genes, relative expression of several components of jasmonic acid signaling was tested under high and low temperatures. Among the four tested JA associated genes in *Sr6* lines, three were downregulated under low temperature except *JA3,* which was upregulated at day 1 (Supplementary Fig. 8B). In *Sr13* lines, *JA1* and *JA3* were upregulated under high temperature at day 1, but all JA genes were downregulated in *Sr13* lines under high temperature at day 3 (Supplementary Fig. 9B). JA related genes were downregulated in *Sr21* lines (Supplementary Fig. 10B).

## RNAseq analysis shows divergence in broader expression patterns between *Sr* lines

Differential expression of most genes (35,000–36,000) was non-significant using our analysis criteria. RNAseq analysis indicated that 57 to 872 genes were upregulated in the NILs compared to parental LMPG at both low and high temperatures (Supplementary Fig. 11, Supplementary Data 3-5). Eight genes uniquely expressed in all three NILs, i.e. four upregulated genes; Aquaporin PIP1 mRNA (*PIP1*), Oxygen-evolving enhancer protein-2 (*OEE-2*), Chlorophyll binding protein (*CBP*) and Alpha-amylase/trypsin inhibitor-like protein (*AAMY*) at day 3 were validated through qPCR analysis (Supplementary Data 1). The qPCR results also validated the results obtained from RNAseq analysis (Supplementary Fig. 12). *OEE-2* and *CBP* were upregulated in LMPG-*Sr6* under low temperature but were downregulated under high temperature. *PIP1* was downregulated under both low and high temperatures. *OEE-2* was downregulated in LMPG-*Sr13* under high and low temperature, whereas it was upregulated under high temperature in LMPG-*Sr21*. *PIP1* was upregulated in LMPG-*Sr13* and LMPG-*Sr21* and downregulated in LMPG-*Sr6*. Additionally, *AAMY* was upregulated under both high and low temperatures in LMPG-*Sr21*. The expression of *CBP* was downregulated under high temperature in LMPG-*Sr6* and LMPG-*Sr21* but was upregulated under low temperature in both lines. This indicates that different genes can follow similar or different expression patterns in *Sr* lines at various time points under low and high temperature conditions.

Most differentially expressed genes (DEGs) in LMPG-*Sr13* and LMPG-*Sr21* followed a similar expression pattern that differed from LMPG-*Sr6*. The number of DEGs for a particular NIL under a certain temperature ranged from 26 to 553 (Supplementary Fig. 13). Most of the transcripts upregulated in LMPG-*Sr6* under low temperature (CR) were distributed across the genome, but with a higher number of genes mapped to chromosomes 2B and 3B on both day 1 and day 3 (Supplementary Fig. 14). Similarly, DEGs mapped to most chromosomes in LMPG-*Sr13* and LMPG-*Sr21*, with some variation where reads tended to map frequently to chromosomes 1A, 2B, and homoeologous group 7 members.

GO analyses were conducted using GeneOntology and ShinyGo (Supplementary Figs. 15–21; Supplementary Data 3–5). Enrichment analysis in *Sr6* lines demonstrated that genes related to defence response to fungal infection, response to biotic stimulus, and various pathogen resistance proteins were expressed under low temperature. These pathways corresponded with the strong upregulation of *PR* genes for *Sr6* at low temperature. On the contrary, significant pathways were not observed for LMPG-*Sr13* under low temperature on day 1. However, when at the high temperature where *Sr13* is most effective, numerous pathways, including ribosomal small subunit assembly, carbon utilization, and photosynthesis were upregulated. Upregulated genes included OEE-2 and Cytochrome C. For LMPG-*Sr21* proton transmembrane transport was upregulated on day 1 at both high and low temperatures. Numerous differences in upregulated pathways were observed at day 3 where various biosynthetic and metabolic processes were upregulated at high temperature, but not at low temperature. Genes upregulated on day 3 included kinase proteins, an ABC transporter, OEE-2, and several membrane trafficking proteins.

## Protein structural comparison and analysis

To further investigate any potential association between temperature-sensitive resistance exhibited by SR6, SR13, and SR21, we performed in silico protein structure modeling using AlphaFold3[38]. Full-length protein structure prediction of all three proteins clearly displayed three subdomains: an N-terminal domain, an NB-ARC domain, and an LRR domain with overall high confidence (Supplementary Figs. 22A and 23). Although all three full length structures were well conserved in protein subdomain composition, the N terminal domains of SR6 and SR21 were significantly different from SR13 that contains a four-helix bundle to comprise a coiled-coil structure (Supplementary Fig. 22B). The N terminal subdomain of SR6 formed by four helices with a BED domain motif insert between a3 and a4 in contrast to the N terminal subdomain of SR21 which appears to form an extra helix in addition to the four helices coiled-coil bundle (Supplementary Fig. 22C). Nevertheless, despite all these distinctive N terminal structures, all three proteins seem to have a highly conserved first three helices bundle and noticeably the a3 helix is always the helix that is physically most adjacent to the LRR domain (Supplementary Fig. 22C). We further compared the zinc finger BED motif from SR6 with other zinc finger BED motifs from other BED domain-containing resistance proteins. All zinc finger BED structures from SR6, YR5a, RPH15, and XA1 highly resembled each other with pairwise Root-Mean-Square Deviation (RMSD) values less than 1 (Supplementary Fig. 24A), despite the rather low sequence similarity (Supplementary Fig. 24B). The WebLogo diagram unsurprisingly displayed a high conservation only for cysteines and histidines, which are predicted to bind to the zinc ions ($Zn^{2+}$) and resemble a "finger" shape (Supplementary Fig. 24C). Overall, we identified no obvious correlation between the resistance temperature sensitivity and the protein structural compositions of SR6, SR13, and SR21.

## Promoter analysis and sub-cellular localization examinations

Cis-regulatory elements play a key role in enhancing, repressing, or precisely regulating gene expression, ensuring that genes function at the right time, location, and under appropriate conditions. To investigate whether specific cis-regulatory elements are involved in temperature-dependent wheat stem rust resistance, we analyzed the promoter regions of *Sr6, Sr13,* and *Sr21*.

After filtering out all known functional cis-regulatory elements from the predicted promoter regions of the three genes, only one temperature-related regulatory element, i.e., the low-temperature responsiveness (LTR) motif CCGAAA, was identified. This element was only present in the *Sr13* promoter region, but absent in *Sr6* and *Sr21* (Supplementary Data 6).

We also performed subcellular localization analyses using the full-length proteins and CC domains plus BED motifs of SR6, SR13, and SR21. When these constructs were expressed in *Nicotiana*

*benthamiana* leaves and incubated at either 18 °C or 25 °C, we found no differences in protein localization between the two temperature conditions or between the different protein constructs (Supplementary Fig. 25).

## Discussion

Mourad et al.[8] identified 32 *Sr6*-associated SNPs that intersected functional gene annotations in Chinese Spring. However, none of them overlapped with the candidate we identified in this study. We isolated *Sr6* using MutRenSeq. After *Yr5/YrSP* and *Yr7*, *Sr6* is the fourth all-stage cloned wheat rust *R* gene containing a BED domain. However, such domain integrations are not uncommon in angiosperms. The rice resistance gene *Xa1* was identified as a BED-containing NLR[40], and recent genomic analyses identified many NLRs that incorporate non-canonical, integrated domains (IDs)[47–49]. The BED domain does appear to be required for resistance, as Marchal et al.[39] noted that a single, induced SNP in the BED domain of *Yr7* led to loss of resistance. Nevertheless, little is known about the functional role of BED domains in immune receptors despite a thorough genome-wide comparative evolutionary analysis of zinc finger BED transcription factor genes in land plants[44]. In Arabidopsis, the non-NB-LRR gene *DAYSLEEPER* encodes a BED domain that was shown to bind DNA[50]. In the context of immunity, IDs are known to act as decoys for pathogen effectors. For example, an integrated WRKY domain in Arabidopsis *R* gene *RRS1-R* binds bacterial effectors that target WRKY transcription factors[51]. It is unclear whether BED domains serve a similar purpose in ETI. However, the identification of a BED domain within the barley leaf rust resistance gene *Rph15* suggests that different rust pathogen species adapted to distinct hosts may have effectors targeting similar transcription factor domains[42]. The BED domain in *Sr6* appeared to be conserved across numerous accessions, not unlike the CC domain, so a potential role in signaling cannot be ruled out.

Our in silico structural modeling and analysis did not show an association between the full-length structures or diverse N-terminal structures of SR6, SR13, SR21, and temperature sensitivity of resistance. *Sr13* encodes a classic CC-NB-LRR, *Sr21* and the *SR9* allelic series encode NLR proteins with a unique N terminus comprised of a bundle of five helices[52], whereas *Sr6* falls into the zinc finger BED domain containing R protein encoding gene subset. Interestingly, we observed that all α3 helices from these proteins were the closest unit to their respective LRR domains. It is consistent with previous studies showing that the EDVID motif in the α3 helix from the CC domain of CNL protein SR35 interacts with its LRR domain to form a conserved EDVID-LRR$^{R-cluster}$ interface. It is considered an evolutionarily conserved stabilization mechanism of CNL resistosomes and is important in mediating resistance-induced cell death[53]. Our observation implies that the zinc finger BED-containing group (SR6, YR5a, RPH15, XA1), SR21, and the SR9 group could potentially share this conserved mechanism. Assuming this is the case, the possible correlation between the temperature sensitivity of the resistance and the distinct N terminal structure of the three proteins in the current study is low.

Transgenic complementation of susceptible Westonia plants with *Sr6* was successful in transferring resistance against *Pgt*. Reduced resistance at high temperatures was also demonstrated in transgenic plants. RNA analysis of the same transgenic plant transitioned between different temperatures may provide enhanced resolution of temperature-dependent expression activity in future studies.

Temperature has been shown to affect the relative abundance of transcript isoforms. Takabatake et al.[54] found that the tobacco (*N. glutinosa*) *N* gene, which encodes an NLR, generates two alternative transcripts: a complete form encoding the full protein, and a truncated form encoding a protein lacking most of the LRR; both transcripts accumulated at 20 °C, a permissive temperature, but not at 30 °C, a non-permissive temperature, in tobacco mosaic virus-inoculated leaves. However, the levels of the complete isoform were always higher just before resistance-related cell death. Like *Sr6*, the *N* gene can be reversibly inactivated, and this phenomenon is maintained in heterologous transgenic systems such as tomato and *N. benthamiana*, indicating a conservation of signaling components[55].

NLRs are not the only class of *R* genes that exhibit temperature dependency. *Yr36*, conferring partial resistance to wheat stripe rust (caused by *P. striiformis* f. sp. *tritici*), is more effective at higher greenhouse temperatures (25-30°C) and encodes a kinase with a START lipid binding domain[56]. Fu et al.[56] found that *Yr36* generates multiple non-functional transcripts with truncations, and the functional isoform encoding the complete protein was upregulated at higher temperatures, mirroring the enhanced resistance seen at such temperatures. Stripe rust infection did not have a significant effect on functional transcript abundance compared to that of temperature. There is growing evidence that alternative splicing is one of the ways in which plants respond to temperature, as found in Arabidopsis with *LATE ELONGATED HYPOCOTYL* (*LHY*), which displayed temperature-associated isoform switching[57]. Similarly, the expression of *Sr21* was enhanced at higher temperatures, and was shown to encode 10 splice isoforms, albeit within the 3'UTR. However, three specific isoforms had increased abundance at 24 °C compared to 16 °C[21]. *Sr13* is also more effective at higher temperatures (25 °C), but unlike *Sr21*, transcript abundance was not affected by temperature[26].

Based on the above, temperature-dependent resistance appears to be mediated primarily by transcriptional and post-transcriptional regulation. However, post-translational regulation has also been implicated; Zhu et al.[18] proposed that temperature sensitivity was mediated by levels of inactive versus active protein conformations at varying temperatures, citing evidence of mutant R proteins that acquired heat stability, allowing them to remain in an active state at usually non-permissive temperatures. Additionally, little is known about the involvement of downstream signaling elements. In the case of *Sr13*, only *PR* gene expression was affected by temperature and not *Sr13* expression per se. Kiraly et al.[58] suggested that in the case of the tobacco N protein, the expression of downstream signaling components was also impaired at high temperatures. Clearly, the disease resistance pathway involves a myriad of regulatory and signaling components, some of which could be influenced to different degrees by temperature.

For both *Sr21* and *Sr13*, upregulation of *PR* genes was observed in resistant plants at higher temperatures only when inoculated with *Pgt*. *PR* genes were not upregulated in susceptible plants. This was further demonstrated by the results of this study, which showed similar upregulation of these genes at higher temperatures. However, shifts in gene expression in response to temperature were less pronounced in *Sr21*-bearing lines. Overall, these results indicate that the expression response of such *PR* genes is determined not by absolute temperature, but by the activity of the *Sr* gene under permissive conditions.

Wheat lines with temperature-sensitive *Sr6*, effective at lower temperatures, displayed largely unique DEGs and pathways compared to wheat lines with *Sr13* and *Sr21*, which are both more effective at higher temperatures. *PR* genes associated with the SA pathway were upregulated in wheat lines with all three genes relative to the controls under low temperatures for *Sr6* and high temperatures for *Sr13* and *Sr21*. The expression of these *PR* genes was much higher in plants with *Sr6* compared to the other two genes. This is consistent with the strong resistance phenotype of *Sr6* compared to the intermediate phenotypes of *Sr13* and *Sr21*. Our study revealed unique pathways that are expressed in wheat lines with *Sr6* compared to *Sr13* and *Sr21* at different temperatures. Differential gene expression analysis revealed unique genes that are expressed during the onset of resistance or susceptibility. Several genes upregulated at low temperature in LMPG-*Sr6* were also upregulated either during susceptibility or at high temperature in *Sr13* and *Sr21* lines. To further explore the details

of these host-pathogen interactions and possible molecular changes, a study of these genes at the protein level will be necessary. Similarly, genes associated with susceptibility require further study along with pathogen virulence factors. Our study highlights key factors to consider for temperature-dependent experiments on *Sr* gene expression: the effect of races, the frequency of alternative transcripts, overall expression, regulation of *PR* genes, and the currently neglected effect of pre-inoculation temperature.

The cloning of *Sr6* adds to the suite of temperature-sensitive, cloned *R* genes in wheat. Further work will help to decipher the possibly shared mechanisms of temperature sensitivity underlying these genes and to what advantage, if any, temperature sensitivity has in plant immunity, keeping in mind that temperature effects will vary between controlled experiments and in the field. Additionally, the cloning of *Sr6* can expedite marker-assisted screening and breeding programs in places where *Sr6* is still effective.

## Methods

### Plant materials for *Sr6* cloning
Seven $M_3$ putative point mutations for *Sr6* in Chinese Spring*5/Red Egyptian 2D (CS/RE 2D substitution line) were used in RenSeq analysis, namely, 3704-6, 3706-7, 3981-4, 4002-6, 4190-6, 5047-4, and 5140-5. $F_3$ lines derived from reciprocal crosses of CS × CS/RE 2D were used in genetic analysis and confirmation of a diagnostic marker developed from the gene candidate. Fifteen seedlings per line were screened with *Pgt* race 21-2,3,7, and resistant plants had IT 0. The Australian cultivar Westonia was used for transformation, and $T_1$ progeny were assessed for validation of the candidate gene.

### Mutant generation
Seeds of two batches (1200 and 1500) of chromosome substitution line CS/RE 2D were treated by 0.5 or 0.6% (v/v) EMS, respectively, and incubated on a shaker for approximately 18 hours[59]. After treatment, the seed was thoroughly rinsed under running water for at least 2 hours. $M_2$ families, each derived from a single spike of an $M_1$ plant, were screened for stem rust response. Individual plants from segregating progenies were grown and progeny-tested. From these, homozygous susceptible mutants and resistant sibling pairs were recovered. Homozygous susceptible point mutants from seven different plants were identified for subsequent *R* gene enrichment and analysis.

### MutRenSeq
High molecular weight (HMW) DNA was extracted from non-infected seedlings of mutant and WT lines of CS/RE 2D using the phenol-chloroform method[60]. Briefly, leaf tissue was homogenized by grinding in liquid nitrogen and lysed in a buffer of 0.3 M EDTA and 0.05 M Tris-HC1 (pH 7.5), with 5% SDS and proteinase K (1 mg/mL), mixed at a ratio of 6.3:1:1. DNA was extracted by mixing lysate with equal volumes of phenol/chloroform/isoamyl alcohol (25:24:1), followed by phase separation, recovery of the aqueous phase, and ethanol precipitation. DNA quality was assessed using a NanoDrop spectrophotometer (Thermo Fisher Scientific, Waltham, MA, USA) and 0.8% agarose gels. Target *R* gene enrichment was performed at Arbor Biosciences (Ann Arbor, MI, USA) following the MYbaits protocol using the custom *Triticeae* RenSeq Bait Library v1 (https://github.com/steuernb/MutantHunter). Library preparation followed TruSeq protocols, and sequencing was performed on a HiSeq 2500 (Illumina) to generate 250 bp paired-end (PE) reads[61]. Subsequent read quality trimming, alignment, and filtering were carried out[62], including SNP calling and candidate identification using the MuTrigo package (https://github.com/TC-Hewitt/MuTrigo). WT re-assembly using reads pooled from all samples (to increase coverage and enhance contiguity) was performed using the BioKanga package v4.4.2 (https://github.com/csiro-crop-informatics/biokanga) to trim raw reads with *biokanga filter*, assemble reads with *biokanga assemb*, and create scaffolds with *biokanga scaffold* using option *-P 1600*, with all other parameters as default[63].

### Confirmation of SNPs and domain prediction
PCR amplification of coding regions was performed on the HMW DNA of mutants and WT. Products were cloned using a TOPO™ XL-2 vector cloning kit (Invitrogen, Mulgrave, VIC, Australia). Plasmid DNA from at least four colonies per product was isolated using an ISOLATE II Plasmid Mini Kit (Bioline, Alexandria, NSW, Australia) and was then Sanger sequenced using primers listed in Supplementary Table 2. Coding sequences and translation were predicted using FGENESH (http://www.softberry.com/berry.phtml), and amino-acid substitutions were confirmed on alignment to the mRNA sequence using CodonCode Aligner v8.0 (https://www.codoncode.com/aligner/). NB-LRR motif annotations were created using NLR-Parser (https://github.com/steuernb/NLR-Parser), and Zfn-BED domain prediction was performed using SMART (http://smart.embl.de) with added PFAM (v8) domains[64].

### *Sr6* transformation
A 8160 bp genomic sequence of *Sr6*, including all introns, 2 kb upstream of 5′UTR and 1 kb downstream of the putative stop codon, encompassing the native promoter and terminator, was synthesized and cloned (Epoch Life Sciences, Missouri City, TX, USA) into *Not*I/*Sgs*I-digested binary vector VecBarIII. Agrobacterium strain AGL1 was used to carry the construct, and the *Sr6* gene was introduced into cv. Westonia using the Agrobacterium-transformation protocol[65] and phosphinothricin as the selective agent. Two independent transgenic plants ($T_0$) carrying the *Sr6* transgene were recovered as confirmed with both PCR markers *Sr6STS1* and *Sr6STS2* (Supplementary Table 2). Seeds from both $T_0$ plants were harvested, and $T_1$ seedlings were grown at low (18°C) and high (25°C) temperatures in growth cabinets along with WT and positive and negative controls. After two weeks, the WT, controls, and $T_1$ plants were inoculated with *Pgt* race 21-0. Phenotyping and photography of leaf samples were performed at 14 days post-inoculation. $T_2$ populations were phenotyped under the same conditions as $T_1$ plants at both temperatures.

### Planting, inoculation, and sample collection for RNAseq
Near-isogenic wheat lines carrying *Sr6*, *Sr13*, and *Sr21*, along with recurrent parent LMPG-6[66] were planted in coarse vermiculite in plastic pots (5 ×5 cm). Twenty seeds of a given line were planted in each pot. The North American wheat stem rust differential set[12] was planted for each treatment. Temperature treatments were conducted under high (25–22 °C) and low (18–15 °C) regimes (15-hr day/9-hr night) in controlled-environment cabinets. Pathogen treatments included spraying with either *Pgt* urediniospores mixed with Soltrol® or only Soltrol as a mock-inoculated treatment. Ten-day-old wheat seedlings were inoculated with urediniospores of *Pgt* isolate 59KS19 (race MCCFC, avirulent to *Sr6*, *Sr13*, and *Sr21*) according to previously described conditions[67]. Twenty plants of each line were included in each temperature and pathogen treatment. The entire experiment was repeated three times with different planting and inoculation dates. RNAseq was performed for each genotype at each temperature and inoculation treatment at two time points: 24 and 72 hours after inoculation. Six primary leaves were collected for RNAseq at each timepoint, temperature, and inoculation treatment. In total, 96 samples were collected and subjected to RNAseq. Surrogate *Pgt* isolate 04KEN156/04 (race TTKSK, avirulent to *Sr13* and *Sr21*) was used for the *Sr6* and *Sr13* inoculations in Fig. 3.

Primary leaf tissue was collected at 24 h and 72 h post inoculation for RNAseq and qPCR experiments. Young leaves were collected in 2 ml flat bottom Eppendorf tubes. All samples were flash frozen in liquid nitrogen and stored at −80 °C for successive experiments.

Seedling infection types on unharvested primary leaves were observed and recorded 14 days post inoculation[68]. Infection types (IT) were rated on a 0–4 scale, with '0' indicating no visible symptoms and '4' being completely susceptible. IT greater than or equal to '3' was considered susceptible, and less than '3' was considered resistant. IT were recorded on 5–6 seedlings of each control and experimental line in each treatment.

### RNA isolation, cDNA library construction, and sequencing

The collected tissues were subjected to RNA extraction using RNeasy® Plant Mini kits (QIAGEN, Chadstone Center, VIC, Australia). On-column DNase digestion (30 min) was performed to remove possible DNA contamination. RNA was eluted from each sample using 50 μl of elution buffer and quality-checked on a Nanodrop ND-1000 spectro-photometer and agarose gel electrophoresis. Pre-screened RNA samples were subjected to TrueSeq and RiboZero library preparation at the University of Minnesota Genomics Center and quality-checked on Bioanalyzer chips (Agilent Technologies). Prepared libraries were first sequenced on a NovaSeq 6000 S1 flow cell to obtain 100 bp PE reads with a target of 35 million reads per sample followed by a second sequencing run on a NovaSeq 6000 S4 flow cell to obtain 150 PE reads with a target of 37 million reads per sample.

The sequence reads were quality-checked using FastQC and quality trimmed to 100 bp PE by removing barcodes and Illumina adapter sequences. An index of high-confidence protein-coding sequences obtained from the IWGSC RefSeq annotation v1.1 was created with kallisto 0.46.1[69]. The transcript sequences were also aligned to this index using kallisto ("quant" command) with 1000 bootstraps of the PE reads to obtain alignment statistics. Alignment counts were further examined for DEGs in the R package edgeR[70], considering log fold change ≥2 and a false discovery rate (FDR) of <0.05 by comparing infected lines to both mock-inoculated near-isogenic lines and inoculated-LMPG (Supplementary Data 3-5). A two-sided statistical test was used with multiple comparisons performed, incorporating a design matrix to account for genotype, treatment, and timepoint effects. Analyses were conducted using edgeR[70]. Furthermore, edgeR implements a general linear model for an exact test under a negative binomial distribution to determine the significance ($p$ values) of differential gene expression[71]. To control the FDR, a multiplicity correction was performed by applying the Benjamini-Hochberg method on the $p$-values[72]. An exon was retained when it was expressed >1 count per million (CPM) in at least three samples. Several DEGs were validated through qPCR. Genes unique to all three lines, along with common genes with similar expression patterns in all three lines, were selected for qPCR.

### RNA analysis and qPCR

For qPCR analysis, RNA extracted from leaf tissue (0, 1, and 3 dpi) was reverse transcribed into cDNA using ProtoScript® II Reverse Transcriptase (NEB). qPCR was performed with TaqMan Gene Expression Master Mix (Applied Biosystems, Foster City, CA) and using an ABI 7300 Real-Time PCR System (Applied Biosystems). Thermal cycling conditions followed the manufacturer's standard protocol. Probe-based IDT primers with a FAM reporter and IBFQ quencher were employed. PCR was carried out under standard cycling conditions for 40 cycles: UNG incubation at 50 °C for 2 minutes, polymerase activation at 95 °C for 15 seconds, followed by denaturation at 95 °C for 15 seconds and annealing/extension at 60 °C for 1 minute. Relative expression was analyzed using $2^{-\Delta\Delta C_T}$ to determine relative quantification[73]. Student's $t$-test was performed for the determination of statistical significance[74]. Fold expression of $Sr$ lines was compared with susceptible LMPG-6 lines at both low and high temperatures.

Four upregulated genes from RNAseq were selected for expression and validation. Additionally, $PR$ genes linked to the SA pathway and genes related to the JA pathway identified from RNAseq were analyzed through qPCR (Supplementary Data 1). Primers for six PR-related and four JA-related genes were tested to investigate their roles in defense across temperature conditions. Applying 18S and GADPH as internal controls, transcript abundance of PR and JA genes in Sr-LMPG and LMPG-6 lines was compared. Resistance gene expression ($Sr6$, $Sr13$, or $Sr21$) for each near-isogenic line was tested on days 1 and 3 under both low and high temperatures, with LMPG-6 as the susceptible control for all qPCR validations.

RT-qPCR was carried out independently on transgenic $Sr6$ material using an alternative method. First, leaf tissues from each sample were frozen in liquid nitrogen or dry ice immediately after sampling; RNA was isolated using a RNeasy® Plant Mini Kit (QIAGEN) according to the manufacturer's protocol, and used in first-strand DNA synthesis in 20 μL reactions using a Superscript® III reverse transcriptase kit (Life Technologies, Mulgrave, VIC, Australia). After the reverse transcript reaction, 3 μL of 10 ng/μL cDNA product was used for qPCR using a C1000 TouchTM thermocycler with the CFX96TM Real-Time System (Bio-Rad). qPCR conditions included an initial denaturation at 95 °C for 3 min; 40 cycles of denaturation at 95 °C for 10 s and annealing/elongation at 60 °C for 30 s, followed by a melt step range of 65–95 °C with increments of 0.5 °C. The wheat housekeeping gene TaCON was used as the reference gene for each qRT-PCR experiment[70]. qPCR primers specific for $Sr6$ (forward: 5'-GTCAATAGCGCCGAGTGTAAG-3', reverse: 5'-GGTCTGATGG CTGAATTACTGG-3') were used to measure relative gene expression using three technical replicates for each sample. δCq mean values were calculated, and standard errors were determined. Gene expression values were log (base 2)-transformed. Boxplots of aggregated relative expression values for each temperature regime and disease phenotype were generated using "geom_boxplot" and "geom_point" functions in R package ggplot2 (v3.3.6). Welch's two-tailed unpaired $t$-tests were performed using the "t.test" function from base R.

RNA for rapid amplification of cDNA ends (RACE) was extracted from leaves of unchallenged, 3-week-old seedlings of CS/RE 2D grown at ambient temperature (20–25 °C) using the RNeasy Plant Mini Kit (Qiagen); 5' and 3' RACE was conducted using a SMARTer RACE 5'/3' Kit (Clontech, Mountain View, CA, USA). RACE products were cloned using a TOPO™ XL-2 vector cloning kit (Invitrogen). Plasmids were isolated from 10 colonies per product using an ISOLATE II Plasmid Mini Kit (Bioline) and Sanger sequenced.

### Gene annotation and pathway analysis

Pathway analysis and biological functions of DEGs were conducted via GeneOntology[75], AgrigoV2[76], and ShinyGo[77]. AgrigoV2 was used to identify the gene networks and pathways associated with the DEGs in different NILs at low or high temperatures. GeneOntology software was used to categorize the DEGs into gene, family, protein, and species groups. The distribution of DEGs according to functional processes was represented in pie charts (Supplementary Fig. 15). ShinyGo was used to plot the DEGs across the chromosomes and calculate the statistical significance of various genomic regions[77]. Graphical enrichment analysis was completed using ShinyGo, where query genes were mapped to the list of genes in particular pathways (Supplementary Figs. 16–21). Fold enrichment was calculated by dividing the percentage of genes in the query list associated with a given pathway by the corresponding percentage of genes in the background. FDR indicated the chances of fold enrichment and was calculated according to the Hypergeometric test[77].

### Primer design and reference comparison

Locus-specific primers were designed manually based on polymorphisms in alignments between $Sr6$ and high-scoring BLAST hits with Chinese Spring IWGSC RefSeq v1.0 (CSv1). DNA was isolated from leaves of $F_3$ lines using the method described in Ellis et al.[78] conducted on a Microlab NIMBUS™ liquid handling robot (Hamilton, Reno, NV).

STS marker screens and gene amplification were carried out using primers listed in Supplementary Table 2. Products were checked by electrophoresis on 1% agarose gels.

Marker *Xwmc453* proximal to *Sr6*[32] was positioned on CSv1 chromosome 2D using BLAST v2.3 (https://www.ncbi.nlm.nih.gov/books/NBK131777/) with the probe sequence retrieved from GrainGenes (https://wheat.pw.usda.gov/GG3/). Homologous *Sr6* sequences were identified from CSv1 and various reference assemblies of the wheat 10+ Genome Project (http://www.10wheatgenomes.com/), listed in Adamski et al.[79], using BLAST v2.9. Full protein sequences of Xa1, Xo1, Rph15, Pm6Sl, Yr5 and Yr7 were aligned with Sr6 using Clustal Omega (https://www.ebi.ac.uk/Tools/msa/clustalo/). The transcript assembly of wheat cv. Avocet R used was the same as that described in Hewitt et al.[62].

### NLR dendrogram construction
NLR protein sequences with an N-terminal coiled-coil domain (CNL class) were taken from the NCBI database. Accession numbers are listed in Supplementary Data 7; 125 sequences were aligned using MUSCLE, and a phylogenetic tree was generated using the UPGMA method in MEGA X (www.megasoftware.net). Distances were computed using the Poisson correction method and are in units of the number of amino acid substitutions per site.

### Protein structural modeling and analysis
Full-length protein structure, ATP and zinc ion binding regions in Fig. 1B were predicted by AlphaFold3 via AlphaFold Server (https://golgi.sandbox.google.com/)[38]. Protein structures in Supplementary Figs. 22 and 23 were predicted by AlphaFold2 using ColabFold (https://colab.research.google.com/github/sokrypton/ColabFold/blob/main/beta/AlphaFold2_advanced.ipynb)[80,81]. Structural and sequence analyses were carried out with PyMOL v2.5.5 and CLC Sequence Viewer v8. Weblogo diagram in Supplementary Fig. 24C was generated by Weblogo (https://weblogo.berkeley.edu/logo.cgi).

### Promoter sequence analysis
For *Sr6*, *Sr13*, and *Sr21*, a 2 kb sequence upstream of the start codon (ATG) was selected as the promoter region for analysis using PlantCARE database (https://bioinformatics.psb.ugent.be/webtools/plantcare/html/)[82]. All cis-acting regulatory elements with known functions were filtered out for further analysis (Supplementary Data 6).

### Sub-cellular localization analysis
The full-length cDNA (without stop codon) of *Sr6*, *Sr13_R1* (Kronos) and *Sr21_R1* (DV92), along with the CC-domain including the BED domain of *Sr6*, the CC-domains of *Sr13_R1* and *Sr21_R1* (each with an additional stop codon) were synthesized by GenScript Biotech (Singapore). Each sequence included an *attB1* and an *attB2* site at 5′ and 3′ ends, respectively. BP recombination reactions were performed using the plasmid containing the sequences and *pDON207*, resulting in a set of entry vectors, including pEntry-Sr6, pEntry-Sr13, pEntry-Sr21, pEntry-Sr6CC, pEntry-Sr13CC, and pEntry-Sr21CC.

Six constructs were generated to examine the sub-cellular localization of the N-terminal domain and the full-length proteins for SR6, SR13, and SR21. The full-length cDNA entry vectors were recombined into the GW-YFP vector (with the yellow fluorescence protein (YFP) at the C-terminus) by LR reaction, generating the constructs Sr6-YFP, Sr13-YFP, and Sr21-YFP. Similarly, the CC-domain entry vectors were recombined into the YFP-GW vector (with YFP at N-terminus) by LR reaction, generating the constructs YFP-Sr6CC, YFP-Sr13CC, and YFP-Sr21CC.

Because Sr13-YFP induced cell death when expressed in *N. benthamiana* leaves, an alternative YFP-Sr13 construct was generated. To create this, the full-length *Sr13* fragment with a stop codon was amplified using the following primers (forward prime: GGGGACAAGTTTGTACAAAAAAGCA and reverse primer: GGGGACCACTTTGTACAAGAAAGCTGGGTCCTTGGAGAAGATCTCACGT) and a plasmid containing *Sr13* from GenScript as a template. The PCR product was cloned into pDON207 by BP reaction, resulting in an entry vector pEntry-Sr13STOP, which was subsequently recombined into YFP-GW by LR reaction, generating the final construct YFP-Sr13.

Plasmids Sr6-YFP, Sr13-YFP, and Sr21-YFP were transformed into GV3101 pMP90 competent cells by electroporation, grown on LB plate containing 25 μg/mL of rifampicin, 15 μg/mL of gentamycin, and 50 μg/mL of carbenicillin. Plasmids YFP-Sr6CC, YFP-Sr13CC, YFP-Sr21CC, and YFP-Sr13 were transformed into GV3103 pMP90 TK competent cells, grown on LB plates containing 25 μg/mL of rifampicin, 15 μg/mL of gentamycin, and 50 μg/mL of kanamycin.

Overnight cultures of transformed Agrobacterium were harvested by centrifugation, and the pellets were resuspended in infiltration buffer containing 1 mM of MES, 1 mM of $MgCl_2$, 200 μM of acetosyringone, pH 5.7. The resuspended cells were incubated at room temperature for at least two hours before infiltration. Prior to infiltration, the Agrobacterium suspensions were adjusted to $OD_{600} = 1.0$. Agrobacterium was infiltrated into leaves of four-week-old *N. benthamiana* plants that had been grown at 24 °C under a 16-hour light/8-hour darkness cycle. The Agrobacterium cultures were infiltrated into either half or whole leaves, and the plants were incubated at either 25 °C or 18 °C under the same light/darkness cycle. Leaf discs were collected at 48 hours post infiltration and mounted on microscope slides for visualization using a Leica SP8 Confocal & Multiphoton Microscope.

### Reporting summary
Further information on research design is available in the Nature Portfolio Reporting Summary linked to this article.

## Data availability
mRNA sequence of *Sr6* is available at NCBI GenBank under accession PP949235. Sequencing reads for MutRenSeq have been deposited at DDBJ/EMBL/GenBank under BioProject PRJNA1127689 of accessions SRX28382285–SRX28382292. Source data are provided with this paper.

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

## Acknowledgements

T.C.H., S.S., R.Mc., E.L., and P.Z. acknowledge support from Grains Research and Development Corporation (GRDC), Australia. J.Z. acknowledges support from an Australian Research Council (ARC) Early Career Industry Fellowship (IE230100282). M.N.R. acknowledges support from the USDA-ARS National Plant Disease Recovery System and a fellowship under the OECD Co-operative Research Program: Biological Resource Management for Sustainable Agricultural Systems. M.R. and P.Z. thank Krista Ristinen, USDA-ARS, for excellent assistance in quantifying stem rust infection phenotypes.

## Author contributions

P.Z., R.M., J.Z., S.S.: generation of mutants, disease phenotyping, and genotyping; T.H.: *Sr6* mutant data analysis and candidate validation. K.S.: qPCR, RNAseq analysis, and GO analysis of NILs; J.Z., C.C.: sub cellular localization analysis; C.C.: genotyping of transgenic plants; D.B.: generation of transgenic plants; P.Z, S.S.: growth, scoring and genotyping of transgenic plants; J.Z., Q.W.: Protein structure prediction and analysis; J.Y.: qPCR analysis of transgenic material; N.U.: created transcript assembly; P.B.: processed RNAseq data; K.S., P.O., M.R.: disease phenotyping of NILs; C.P.: provided Landmark sequence assembly before its publication; E.L., M.R., R.M., P.Z. designed and supervised the study; T.H., K.S., J.Z., M.R., R.M., E.L., P.Z. drafted the manuscript and all co-authors provided edits.

## Competing interests

The authors declare no competing interests.
