## [Peer Review file · Nature Communications]

Divergent molecular pathways govern temperature-dependent wheat stem rust resistance genes

Corresponding Author: Professor Peng Zhang

Version 0:

Reviewer comments:

Reviewer #1

(Remarks to the Author)

In this manuscript, Hewitt et al. identify the Sr6 CC-BED-NLR gene of wheat and characterise differences in temperature-dependent immune responses regulated by Sr6 versus other wheat stem rust NLRs. They perform RNAseq analyses and describe major differences between Sr6-mediated immunity compared to Sr13 or Sr21-mediated immunity. The breakthrough here appears to be the cloning of Sr6 and the confirmation of its temperature-dependent deployment in wheat. The authors describe differences in immunity at different temperatures. Major and minor concerns are below.

Major Concerns:

1. Conserved BED domain: Minimal support for the statement/title in Line 158 that BED is conserved. The others provide a short alignment showing minimal overlap between BED domains. Further analysis of BED-NLR ancestry and similarity using a greater number of CC-BED-NLRs would be required to support the statement that Sr6 is diverged but contains a conserved BED. Lines 299-321 do a better job at characterizing BED domain structure and sequence so consider toning down the earlier section and making the point later on in the manuscript.
2. Quantification of infection phenotypes: Data is presented with single images and the authors would be better served by providing some degree of quantitation to support their claims, especially when comparing resistance across different conditions/temperatures/transgenic lines etc.
3. RNA-seq data: As far as I can tell, the RNAseq data presented in the supplementary data are not quantitative and are therefore not as informative as they could be. It is essential to provide measures and statistics here. To what degree were these genes differentially expressed (Log2FC), and how reliable is this information (Pvalues)?
4. Data Availability: structure models and NLR sequences used to generate them should be provided, either in the supplementary or through an online platform like FigShare or Zenodo.

Minor Concerns:

- Line 73: mediated is more appropriate than 'induced' here.
- Figure 1: in (A) remove the speculative "translation efficiency?" statement. In (B) include prediction metrics to give an idea of how well the NLR was modelled by AF3.
- Please show the 'data not shown'.
- What are JA genes? Please define what markers these are.

Reviewer #2

(Remarks to the Author)

Remarks to the Author:

Title: Divergent molecular pathways govern temperature-dependent wheat stem rust resistance genes Sr6, Sr13 and Sr21

Authors: Zhang et al.

Ms #: Nature Communications: NCOMMS-24-40169

Wheat stem rust, caused by the fungal pathogen *Puccinia graminis* f. sp. *tritici* (Pgt), used to severely threaten world wheat production before the 1950s. The outbreak of the disease had been well controlled due to the deployment of resistant genes in wheat cultivars and the removal of pathogen hosts. However, since 1999, when a novel virulent strain, Ug99, was discovered in Uganda, stem rust threats to wheat began to be highly regarded. Ug99 and its virulent variants have overcome widely deployed disease resistance genes and the majority of cataloged Sr genes. Cloning Sr genes helps to understand the mechanism of interactions between plant and pathogen Pgt, thus accelerating the exploration of novel resistance genes and wheat breeding for resistance to stem rust. In this manuscript, the authors used mutagenesis and resistance gene enrichment and sequencing (MutRenSeq) to clone a stem rust resistance gene, Sr6, encoding an NLR protein with an integrated BED domain. It is the first characterized BED-NLR conferring resistance to wheat stem rust, providing a new member of crop BED-NLR for understanding the mechanism of BED-NLR-mediated resistance to diverse pathogens.

Major Concerns:

1. Although Sr6 cloning contributes to understanding the mechanism of wheat resistance to stem rust, it is susceptible to Ug99 and its virulent variants, so its potential for application in modern wheat breeding programs is very limited.
2. Transgenic complementation is a key proof of candidate gene function. However, analysis of transgene expression levels obviously conflicts with phenotype assessment results and the ratio of resistant plants to susceptible plants. The authors claimed that although all transgenic plants in T1 families, either resistant or susceptible, were positive for Sr6. They identified transgene expression levels, found that the expression level of susceptible plants is zero (exactly zero?), and indicated that the position effect of Sr6 transgene may lead to its non-expression and thus susceptibility. However, as displaying in Fig 4, some transgenic resistant plants also had Sr6 expression levels very close to zero at 18°C, and around half of the susceptible plants did have expression levels similar to that of the resistant plants at 25°C. Gene expression level obviously conflicts phenotype and genotype of transgenic plants, and it is not convincing to explain it using the transgene position effect. In addition, expression levels of resistant T1 plants varied significantly. Is it the result of one T1 family, or two T1 families together? Given the importance of transgenic complementation for gene cloning, it is suggested that genetic transformation be carried out again to produce more transgenic events. The likelihood of two independent T1 families both positive for transgene having non-expressed copies is low. Transgenic copy number analysis is required on each T1 plant to resolve conflicts. The ratio of resistant plants to susceptible plants and the difference in expression levels of resistant plants to susceptible plants should be recalculated for each T1 family.
3. The authors cloned Sr6 gene, analyzing molecular pathways involved in stem rust resistance mediated by temperature-dependent wheat stem rust resistance genes Sr6, Sr13, and Sr21. By differential gene expression analysis using near-isogenic wheat lines inoculated with Pgt at varying temperatures, they found that genes upregulated in the low-temperature-effective Sr6 response differed significantly from those upregulated in the high-temperature-effective Sr13 and Sr21 responses. However, this section lacks connection to Sr6 cloning. Suggest adding analyses of native promoter elements, subcellular location and expression of different protein structures and domains at two different temperatures.

Minor concerns:

1. Comprehensive formatting changes are required, particularly on lines 193 (*Agrobacterium*), 337 and 340 (*Arabidopsis*): italics; 189 & 263 (spaces between words); and 257 (periods).
2. The authors introduced Fig. 2 followed by Fig. 4 in the manuscript, the sequence of Fig 3 and 4 should be rearranged.
3. In Fig 3 c, d, it is very difficult to distinguish the difference in symptoms between low temperature and high temperature, not supporting that Sr13 and Sr21 are temperature sensitive.
4. In Fig 4, the phenotype ratio for transgenic plants is consistent at 3:1, although all are resistant. The segregation ratio cannot be calculated with the combined segregation data from two independent transgenic lines. It should be calculated separately. Some resistant plants exhibit similar expression levels as susceptible ones, particularly at 25°C where resistance decreases while susceptibility increases, how to explain this? However, although the expression level of all susceptible plants in Fig 2 is zero, some resistance plants also had zero Sr6 expression levels. How to explain it? Each plant in Fig 4 should have a corresponding expression level. Explain why single transgenic plants with the same or very similar qPCR relative expression (RE) values of Sr6 responded very differently to Pgt? Considering that two transgenic plants may result from two independent regeneration, analyses of transgenic families in Lines 209-226 should be conducted independently.
5. The authors claimed that Sr6 is upregulated at low temperatures and downregulated at high temperatures. The Sr6 promoter sequence should be examined for temperature sensitive elements to further understand its temperature dependence.
6. In plant materials and methods, provide description how to perform qPCR validation of pathogenesis related (PR) genes (PR1, PR2, PR3, PR4, PR5 and PR9) and JA genes (JA1, JA3, JA4 and JA6). Was LMPG also used to do qPCR

validation? Otherwise in Supplementary Fig.7.2,3,4, how to compare PR and JA genes with those in LMPG (SR6 vs LMPG)?

7. Supplementary Fig.7.5, the authors should add data for RNASEQ HRD3.

8. Line 114 : "No SNP was detected in mutant 3981-4,.....". However, Figure 1A shows that 3981-4 has SNPS in the NB-ARC domain.

9. Line 126-127 : This statement in the text (Lines 114-135) is inconsistent with Fig 1A, where 8 mutation sites on 6 mutants, including 6 missense mutations and 1 premature termination mutation indicated, and 1 nonsense mutation.

10.Lines 191-193: Suggest amplifying the Sr6 sequence using Westonia to confirm whether it carries Sr6 or not.

11.Lines 194-204: Two successful T0 transformants were generated, probably with different copies of transgenes. Thus, the segregation ratio of resistant T1 plants to susceptible T1 plants in each T1 family should be calculated separately. Whether the ratio of susceptible plants at high temperature comports is higher than at low temperature requires statistical analysis.

12. Lines 209-226: Since all susceptible transgenic plants are Sr6 positive, each susceptible plant in a T1 family should have a lower expression level than those resistant plants at the same temperature. It is inappropriate to compare mean expression levels of resistant plants with those of susceptible plants using two independent transgenic T1 families.

13. Lines 241-259: "Biotic stress genes show differential expression between Sr lines at high and low temperatures." Why choose JA and SA pathway genes? Provide evidence that JA and SA pathways are involved in these three genes' resistance. Suggest using RNA-seq data to identify differences in biotic stress-related gene expression between Sr lines at high and low temperatures.

14. Line 258: "JA genes were downregulated in Sr13 lines on all other conditions." What other conditions mean? Are there other treatments besides temperature?

15. Line 276-277: Lack of BCP expression pattern in LMPG-Sr13. The conclusion is inappropriate.

16. Line 293, "Carton"? should "carbon"

17.Lines 322-324, 347-349: suggest to add experiments on temperature affects on subcellular location, expression, protein stability of Sr6, Sr13, Sr21, especially N-terminal structures in Nicotiana tabacum leaf.

18.Supplementary Fig. 1: The nucleotide C changed to T in mutant 3704-6, but it is not shown in Fig. 1

Reviewer #3

(Remarks to the Author)

Reviewer #4

(Remarks to the Author)

This manuscript investigates the temperature-dependent stem rust resistance genes in wheat, which is an interesting topic. The authors cloned the wheat Sr6 resistance gene, which confers resistance under lower temperature. This is in contrast to Sr13 and Sr21, which confer stronger resistance under higher ambient temperature. Furthermore, RNAseq and protein structure analysis were performed to uncover possible mechanisms of the temperature dependency of Sr-mediated resistance. Cloning of Sr6 gene is quite nice and the topic is very interesting. However, I feel the data quality and presentation of the study need to be improved. In addition, RNAseq and Sr protein structure prediction do not really tell much about the underlying mechanism. The authors should develop other strategies to tackle this question. Some specific comments are as below.

1. Are results of Figure 2 and 4 based on T1 plants from only one T0 parent? This is likely insufficient. The authors should use more independent lines.

2. Figure 4, many important information is missing in the figure. The authors should show the disease phenotypes and corresponding Sr6 gene expression of each line in this experiment. The error bars are quite large. Were all the lines tested in one experimental repeat or multiple experimental repeats? Were all T1 plants from two T0 parent lines used in the experiment?

3. Line 233, "Sr6 was upregulated at low temperature (Supplementary Fig. 7.1)". This seems inconsistent with results in Figure 4.

4. Figure 2 and 3, I suggest adding quantitative data showing the severity of disease/fungal multiplication, in addition to pictures of symptoms.

Version 1:

Reviewer comments:

Reviewer #1

(Remarks to the Author)

The authors addressed my main concerns.

Reviewer #2

(Remarks to the Author)

The authors have addressed all previous concerns in a point-by-point manner and provided additional experimental evidence as requested. The manuscript has undergone comprehensive revision with corrections made to formatting errors highlighted in the initial review. However, residual formatting inconsistencies persist in the following sections, such as (1) Lines 203 , 700–716 , 991: Incomplete italicization of genus name “Agrobacterium” (correct: italicized Agrobacterium); (2) Lines 216–219: Improper gene symbol formatting for “Bar” (correct: italicized Bar). No additional scientific concerns were identified.

Reviewer #3

(Remarks to the Author)

Reviewer #4

(Remarks to the Author)

The manuscript is improved, but the mechanisms underlying the temperature-dependent resistance of Sr genes are still elusive. Suggest modifying/tuning down the title of the manuscript-“divergent molecular pathways” are still not clear.

Response to the reviewers' comments

We thank the reviewers for their suggestions/comments, which were incorporated in the revised manuscript. The responses to the reviewers' comments are in blue, whereas the revised text from the paper is marked in red.

Reviewer #1 (Remarks to the Author):

In this manuscript, Hewitt et al. identify the Sr6 CC-BED-NLR gene of wheat and characterise differences in temperature-dependent immune responses regulated by Sr6 versus other wheat stem rust NLRs. They perform RNAseq analyses and describe major differences between Sr6-mediated immunity compared to Sr13 or Sr21-mediated immunity. The breakthrough here appears to be the cloning of Sr6 and the confirmation of its temperature-dependent deployment in wheat. The authors describe differences in immunity at different temperatures. Major and minor concerns are below.

Major Concerns:

1. Conserved BED domain: Minimal support for the statement/title in Line 158 that BED is conserved. The others provide a short alignment showing minimal overlap between BED domains. Further analysis of BED-NLR ancestry and similarity using a greater number of CC-BED-NLRs would be required to support the statement that Sr6 is diverged but contains a conserved BED. Lines 299-321 do a better job at characterizing BED domain structure and sequence so consider toning down the earlier section and making the point later on in the manuscript.

We have updated the relevant sections (Lines 165-166, Lines 173-183, Line 638) and Supplementary Fig. 4 to include additional BED-NLR sequences, highlighting potential conservation in overall domain arrangement and specific residues within the BED domain.

2. Quantification of infection phenotypes: Data is presented with single images and the authors would be better served by providing some degree of quantitation to support their claims, especially when comparing resistance across different conditions/temperatures/transgenic lines etc.

Thank you for this suggestion! We quantified the infection phenotypes and provided appropriate statistics in revised Figures 2, 3, and Supplementary Fig. 6.

3. RNA-seq data: As far as I can tell, the RNAseq data presented in the supplementary data are not quantitative and are therefore not as informative as they could be. It is essential to provide measures and statistics here. To what degree were these genes differentially expressed (Log2FC), and how reliable is this information (Pvalues)?

Thank you for raising this limitation of the available data! We added Log2FC and P values for the RNAseq data in Supplementary Tables 2.5-2.8, 3.5-3.8, and 4.5-4.8. The genes were differentially expressed with P values and FDR values ranging from 0.0002 to 1.46E-23 and from 0.05 to 3.31E-19, respectively.

4. Data Availability: structure models and NLR sequences used to generate them should be provided, either in the supplementary or through an online platform like FigShare or Zenodo.

The structure models and NLR sequences used are provided in Supplementary Data 1.

Minor Concerns:

- Line 73: mediated is more appropriate than 'induced' here.

The suggested change has been made.

- Figure 1: in (A) remove the speculative "translation efficiency?" statement.

Removed as suggested.

In (B) include prediction metrics to give an idea of how well the NLR was modelled by AF3.

The prediction metrics information is included in the pdb files and are provided in Supplementary Data 1.

- Please show the 'data not shown'.

Gel images have been added as Supplementary Figures 2D, 2E.

- What are JA genes? Please define what markers these are.

We added the description of the JA genes in Supplementary Fig. 7.2 and Supplementary Table 8, and clarified these genes in the revised manuscript in Lines 261-264.

Reviewer #2 (Remarks to the Author):

Remarks to the Author:

Title: Divergent molecular pathways govern temperature-dependent wheat stem rust resistance genes Sr6, Sr13 and Sr21

Authors: Zhang et al.

Ms #: Nature Communications: NCOMMS-24-40169

Wheat stem rust, caused by the fungal pathogen *Puccinia graminis* f. sp. *tritici* (Pgt), used to severely threaten world wheat production before the 1950s. The outbreak of the disease had been well controlled due to the deployment of resistant genes in wheat cultivars and the removal of pathogen hosts. However, since 1999, when a novel virulent strain, Ug99, was discovered in Uganda, stem rust threats to wheat began to be highly regarded. Ug99 and its virulent variants have overcome widely deployed disease resistance genes and the majority of cataloged Sr genes. Cloning Sr genes helps to understand the mechanism of interactions between plant and pathogen Pgt, thus accelerating the exploration of novel resistance genes and wheat breeding for resistance to stem rust. In this manuscript, the authors used mutagenesis and resistance gene enrichment and sequencing (MutRenSeq) to clone a stem rust resistance gene, Sr6, encoding an NLR protein with an integrated BED domain. It is the first characterized BED-NLR conferring resistance to wheat stem rust, providing a new member of crop BED-NLR for understanding the mechanism of BED-NLR-mediated resistance to diverse pathogens.

Major Concerns:

1. Although Sr6 cloning contributes to understanding the mechanism of wheat resistance to stem rust, it is susceptible to Ug99 and its virulent variants, so its potential for application in modern wheat breeding programs is very limited.

One of the main aims of our study was to investigate the temperature sensitivity of Sr6 to advance our understanding of the biology and role of temperature sensitivity in crop immunity. We agree with the reviewer that the virulent strains against Sr6 are common, making Sr6 currently less widely effective. However, it continues to be selected for in wheat breeding and is present in some varieties. It provides resistance against the current stem rust pathogen population in the United States and, when combined with other resistance genes, may contribute to overall resistance in North America and India.

2. Transgenic complementation is a key proof of candidate gene function. However, analysis of transgene expression levels obviously conflicts with phenotype assessment results and the ratio of resistant plants to susceptible plants. The authors claimed that although all transgenic plants in T1 families, either resistant or susceptible, were positive for Sr6. They identified transgene expression levels, found that the expression level of susceptible plants is zero (exactly zero?), and indicated that the position effect of Sr6 transgene may lead to its non-expression and thus susceptibility. However, as displaying in Fig 4, some transgenic resistant plants also had Sr6 expression levels very close to zero at 18°C, and around half of the susceptible plants did have expression levels similar to that of the resistant plants at 25°C. Gene expression level obviously conflicts phenotype and genotype of transgenic plants, and it is not convincing to

explain it using the transgene position effect. In addition, expression levels of resistant T1 plants varied significantly. Is it the result of one T1 family, or two T1 families together? Given the importance of transgenic complementation for gene cloning, it is suggested that genetic transformation be carried out again to produce more transgenic events. The likelihood of two independent T1 families both positive for transgene having non-expressed copies is low. Transgenic copy number analysis is required on each T1 plant to resolve conflicts. The ratio of resistant plants to susceptible plants and the difference in expression levels of resistant plants to susceptible plants should be recalculated for each T1 family.

Thank you for the critical review on this section!

We note in the companion paper by Wang et al., who report 97 independent mutants all with non-synonymous or missense nucleotide changes in the gene identical to the one reported as *Sr6* in this manuscript. There is no ambiguity that both groups (Hewitt et al. and Wang et al.) are dealing with the encoding *Sr6* gene. In this instance, the transgenic complementation is primarily a validation exercise. We undertook many transformation experiments with 294 explants in the wheat genotypes of Westonia and Mace. Only two survived beyond the callus phase as shoots. We have never encountered such low regeneration efficiencies with other *Sr* gene constructs we published in the past. It is unclear if the poor survival rate was due to intrinsic effects of the *Sr6* transgene construct.

From the two independent T0 plants, they were subsequently developed into T1 and T2 lines. Because the T1 plants used in the previous experiments were not retained, we used different T1 plants to repeat the experiments. For T1 plants derived from T0 PC326-1, at low temperature, the segregation of resistant : susceptible plants was 30:8, which fits a single gene segregation ratio ($\chi^2_{3:1}=0.316$, $p=0.574$). Similarly, for those derived from T0 PC326-2, at low temperature, the segregation was 19:11, which also fits a single gene segregation ratio ($\chi^2_{3:1}=2.178$, $p=0.140$). At high temperature (this time we used facilities that had better temperature control), all T1 plants were susceptible, confirming that *Sr6* is not effective under high temperature conditions.

We used speed breeding growing conditions and obtained T2 seeds. For T2s derived from T1 of PC326-1, segregation of homozygous resistant : segregating : homozygous susceptible at low temperature was 13:15:8 (two resistant T1 plants did not survive), consistent with a single gene segregation ratio ($\chi^2_{1:2:1}=2.389$, $p=0.3029$). Similarly, for those derived from T2 of PC326-2, segregation for homozygous resistant : segregating : homozygous susceptible at low temperature was 5:13:11 (one resistant T1 plant did not survive), again consistent with a single gene segregation ratio ($\chi^2_{1:2:1}=2.793$, $p=0.2474$). At high temperature, all T2 plants were susceptible.

PCR amplification using the STS marker produced variable results with Westonia and variability in the T1 plants. A careful examination of the available wheat pangenome, including reference lines with pedigrees related to Westonia revealed diagnostic SNPs and indels in the primer binding site of genotypes with and without *Sr6*. Due to these inconsistencies, we opted to genotype the new batch of T1 plants and their T2 progeny using the *Bar* gene, a selectable marker present in the *Sr6* transgene construct. Without exception, the presence of *Bar* was 100% associated with the *Sr6* resistant phenotype. Resistant T1 plants showed *Bar* gene amplification, whereas susceptible plants did not. Similarly, homozygous resistant and segregating T2 lines exhibited *Bar* gene amplification, whereas homozygous susceptible lines did not.

Our new results clearly demonstrated transfer of not only the resistance phenotype, but also temperature sensitivity.

3. The authors cloned *Sr6* gene, analyzing molecular pathways involved in stem rust resistance mediated by temperature-dependent wheat stem rust resistance genes *Sr6*, *Sr13*, and *Sr21*. By differential gene expression analysis using near-isogenic wheat lines inoculated with Pgt at varying temperatures, they found that genes upregulated in the low-temperature-effective *Sr6* response differed significantly from those upregulated in the high-temperature-effective *Sr13* and *Sr21* responses. However, this section lacks connection to *Sr6* cloning. Suggest adding analyses of native promoter elements, subcellular location and expression of different protein structures and domains at two different temperatures.

The native promoter element analysis results of *Sr6*, *Sr13*, and *Sr21* have been added in Lines 339-347, and in Supplementary Table 9. The results of the subcellular localization for all three proteins in full length and CC domain at two different temperatures were also added to the revised manuscript, Lines 348-352, and in Supplementary Figure 9.

Minor concerns:

1. Comprehensive formatting changes are required, particularly on lines 193 (*Agrobacterium*), 337 and 340 (*Arabidopsis*): italics; 189 & 263 (spaces between words); and 257 (periods).

We have gone through the manuscript and made changes. However, we did not italicise *Agrobacterium* and *Arabidopsis* because italics should be used only when the names are used in a taxonomic sense and in these cases they are not.

2. The authors introduced Fig. 2 followed by Fig. 4 in the manuscript, the sequence of Fig 3 and 4 should be rearranged.

In the revised manuscript, we no longer have Fig. 4.

3. In Fig 3 c, d, it is very difficult to distinguish the difference in symptoms between low temperature and high temperature, not supporting that Sr13 and Sr21 are temperature sensitive.

We performed additional phenotypic assays and updated Figure 3, clearly documenting the temperature sensitivity of *Sr13* and *Sr21* with statistics based on quantitative measure of pustule size.

4. In Fig 4, the phenotype ratio for transgenic plants is consistent at 3:1, although all are resistant. The segregation ratio cannot be calculated with the combined segregation data from two independent transgenic lines. It should be calculated separately. Some resistant plants exhibit similar expression levels as susceptible ones, particularly at 25°C where resistance decreases while susceptibility increases, how to explain this? However, although the expression level of all susceptible plants in Fig 2 is zero, some resistance plants also had zero Sr6 expression levels. How to explain it? Each plant in Fig 4 should have a corresponding expression level. Explain why single transgenic plants with the same or very similar qPCR relative expression (RE) values of Sr6 responded very differently to Pgt? Considering that two transgenic plants may result from two independent regeneration, analyses of transgenic families in Lines 209-226 should be conducted independently.

Replies to most of these points are addressed in “Major concerns: 2”. In the revised manuscript, we conducted statistical analysis separately on populations from two independent transgenic lines. For the new experiment, we used the facilities with better temperature control, resulting in a much clearer temperature effect than previously, i.e., plants carrying the transgene were resistant and those without it were susceptible. This was further supported by testing with the selectable marker for the Bar gene, which was part of the *Sr6* transgene construct. Given these results, additional expression studies were considered unnecessary.

5. The authors claimed that Sr6 is upregulated at low temperatures and downregulated at high temperatures. The Sr6 promoter sequence should be examined for temperature sensitive elements to further understand its temperature dependence.

This was addressed in “Major concerns: 3”.

6. In plant materials and methods, provide description how to perform qPCR validation of pathogenesis related (PR) genes (PR1, PR2, PR3, PR4, PR5 and PR9) and JA genes (JA1, JA3, JA4 and JA6). Was LMPG also used to do qPCR validation? Otherwise in Supplementary Fig.7.2,3,4, how to compare PR and JA genes with those in LMPG (SR6 vs LMPG)?

Thank you for raising this limitation of the methods and the problem with interpreting results in Supplementary Fig. 7.2, 7.3, and 7.4. In Lines 578-586 of the revised manuscript, we have added the methods for performing qPCR validation of PR and JA genes. We used LMPG as part of the qPCR validation. This was clarified in Lines 585-586 of the revised manuscript and in the legends of Supplementary Fig. 7.2, 7.3, and 7.4.

7. Supplementary Fig.7.5, the authors should add data for RNASEQ HRD3.

Thank you for identifying this omission! We have added the RNASEQ results for HRD3 in Supplementary Fig. 7.5.

8. Line 114: "No SNP was detected in mutant 3981-4,.....". However, Figure 1A shows that 3981-4 has SNPS in the NB-ARC domain.

Thank you very much for noticing this error! We mislabelled "3981-4" for "3704-6". This has been corrected in Fig. 1A. A SNP was later identified in 3981-4 within the 5'UTR using Sanger sequencing. The change was made in the revised manuscript in Lines 118-119.

9. Line 126-127: This statement in the text (Lines 114-135) is inconsistent with Fig 1A, where 8 mutation sites on 6 mutants, including 6 missense mutations and 1 premature termination mutation indicated, and 1 nonsense mutation.

In the original Fig. 1A, there was a mislabelling error, which was corrected in the revised manuscript. Here, we discussed only the six mutants with SNPs identified from the RenSeq reads, excluding the 7th mutant (3981-4), whose SNP was identified using RACE.

In Lines 128-131, we changed to "Each of the six mutants contained at least one nonsynonymous SNP, with five resulting in missense mutations (mutants 3704-6, 3706-7, 4002-6, 4190-6 and 5047-4) and one resulting in a nonsense mutation (mutant 5140-5)."

10.Lines 191-193: Suggest amplifying the Sr6 sequence using Westonia to confirm whether it carries Sr6 or not.

We used Sr6 marker Sr6STS1 and could not amplify any fragment from Westonia stocks held by CSIRO and the University of Sydney. We suspect that the seed source of Westonia in Park et al. (2009) may not be the same as used in the current study.

11.Lines 194-204: Two successful T0 transformants were generated, probably with different copies of transgenes. Thus, the segregation ratio of resistant T1 plants to susceptible T1 plants in each T1 family should be calculated separately. Whether the

ratio of susceptible plants at high temperature comports is higher than at low temperature requires statistical analysis.

Thank you for your suggestions! This is addressed in “Major concerns: 2”.

12. Lines 209-226: Since all susceptible transgenic plants are Sr6 positive, each susceptible plant in a T1 family should have a lower expression level than those resistant plants at the same temperature. It is inappropriate to compare mean expression levels of resistant plants with those of susceptible plants using two independent transgenic T1 families.

Thank you for your suggestions! We addressed this in “Minor concerns: 4”.

13. Lines 241-259: "Biotic stress genes show differential expression between Sr lines at high and low temperatures." Why choose JA and SA pathway genes? Provide evidence that JA and SA pathways are involved in these three genes' resistance. Suggest using RNA-seq data to identify differences in biotic stress-related gene expression between Sr lines at high and low temperatures.

Previous studies demonstrated involvement of salicylic acid (SA) and jasmonic acid (JA)-responsive PR genes in the function of *Sr13* and *Sr21*^{21,26}. This was clarified in the text of the revised manuscript in Lines 247-255. RNAseq data available in Supplementary Figures 7.2, 7.3, and 7.4 demonstrate the downregulation of JA-associated genes in near-isogenic lines relative to LMPG-6 in environments in which the respective genes were effective.

14. Line 258: "JA genes were downregulated in Sr13 lines on all other conditions." What other conditions mean? Are there other treatments besides temperature?

Thank you for pointing this out! We have clarified the sentence in the revised manuscript Line 269.

15. Line 276-277: Lack of BCP expression pattern in LMPG-Sr13. The conclusion is inappropriate.

Thank you for raising this point! We clarified the lack of an expression pattern for CBP in the revised manuscript Lines 286-290.

16. Line 293, "Carton"? should "carbon"

Thank you! We have fixed the typo.

17. Lines 322-324, 347-349: suggest to add experiments on temperature affects on

subcellular location, expression, protein stability of Sr6, Sr13, Sr21, especially N-terminal structures in *Nicotiana tabacum* leaf.

Addressed in “Major concern” question 3.

18. Supplementary Fig. 1: The nucleotide C changed to T in mutant 3704-6, but it is not shown in Fig. 1

There was a labelling error in Fig. 1A, and has been fixed.

Reviewer #3 (Remarks to the Author):

Reviewer #4 (Remarks to the Author):

This manuscript investigates the temperature-dependent stem rust resistance genes in wheat, which is an interesting topic. The authors cloned the wheat Sr6 resistance gene, which confers resistance under lower temperature. This is in contrast to Sr13 and Sr21, which confer stronger resistance under higher ambient temperature. Furthermore, RNAseq and protein structure analysis were performed to uncover possible mechanisms of the temperature dependency of Sr-mediated resistance. Cloning of Sr6 gene is quite nice and the topic is very interesting. However, I feel the data quality and presentation of the study need to be improved. In addition, RNAseq and Sr protein structure prediction do not really tell much about the underlying mechanism. The authors should develop other strategies to tackle this question. Some specific comments are as below.

1. Are results of Figure 2 and 4 based on T1 plants from only one T0 parent? This is likely insufficient. The authors should use more independent lines.

We had two independent T0 plants. For a detailed response, please see response to Reviewer #2, Major concerns: 2.

2. Figure 4, many important information is missing in the figure. The authors should show the disease phenotypes and corresponding Sr6 gene expression of each line in this experiment. The error bars are quite large. Were all the lines tested in one

experimental repeat or multiple experimental repeats? Were all T1 plants from two T0 parent lines used in the experiment?

We re-performed the phenotyping and genotyping on a new batch of T1 and T2 materials under much better temperature control and obtained much clearer and more consistent results. For details, please refer to the response to Reviewer #2, Major concerns: 2 and Minor concerns: 4.

3. Line 233, "Sr6 was upregulated at low temperature (Supplementary Fig. 7.1)". This seems inconsistent with results in Figure 4.

We acknowledged that the temperature control was not ideal when we performed the experiment for Fig. 4. The results in Supplementary Fig. 7.1 were correct.

4. Figure 2 and 3, I suggest adding quantitative data showing the severity of disease/fungal multiplication, in addition to pictures of symptoms.

We have added the quantitative data in Fig. 2, 3, and Supplementary Fig. 6.

Response to the reviewers' comments

We thank the reviewers for their suggestions/comments. The responses to the reviewers' comments are in blue.

REVIEWERS' COMMENTS

Reviewer #1 (Remarks to the Author):

The authors addressed my main concerns.

Thank you very much!

Reviewer #2 (Remarks to the Author):

The authors have addressed all previous concerns in a point-by-point manner and provided additional experimental evidence as requested. The manuscript has undergone comprehensive revision with corrections made to formatting errors highlighted in the initial review. However, residual formatting inconsistencies persist in the following sections, such as (1) Lines 203, 700 – 716, 991: Incomplete italicization of genus name “*Agrobacterium*” (correct: italicized *Agrobacterium*); (2) Lines 216 – 219: Improper gene symbol formatting for “*Bar*” (correct: italicized *Bar*).

No additional scientific concerns were identified.

Thank you very much for your comments/suggestions! We have italicized “*Bar*”. However, we did not italicise *Agrobacterium* because italics should be used only when the names are use in a taxonomic sense (e.g. *Agrobacterium tumefaciens*) and in these cases they are not.

Reviewer #3 (Remarks to the Author):

Thank you!

Reviewer #4 (Remarks to the Author):

The manuscript is improved, but the mechanisms underlying the temperature-dependent resistance of Sr genes are still elusive. Suggest modifying/tuning down the title of the manuscript-“divergent molecular pathways” are still not clear.

We revised the title. Our future follow-up study is on the elucidation of the mechanisms underlying the temperature-dependent resistance of Sr genes.